# Impact of CPAP Therapy on the Autonomic Nervous System

**DOI:** 10.3390/biomedicines11123210

**Published:** 2023-12-03

**Authors:** Tea Friščić, Domagoj Vidović, Igor Alfirević, Edvard Galić

**Affiliations:** 1Department of Cardiovascular Diseases, Clinical Hospital Sveti Duh, 10000 Zagreb, Croatia; alfirevic.igor@gmail.com (I.A.); edvardgalic1@gmail.com (E.G.); 2University Psychiatric Hospital Vrapče, 10000 Zagreb, Croatia; domagoj.vidovic@bolnica-vrapce.hr; 3Faculty of Croatian Studies, University of Zagreb, 10000 Zagreb, Croatia; 4Libertas international University, 10000 Zagreb, Croatia; 5School of Medicine, University of Zagreb, 10000 Zagreb, Croatia

**Keywords:** obstructive sleep apnea, continuous positive airway pressure, autonomic nervous system, heart rate variability, ambulatory blood pressure monitoring

## Abstract

Obstructive sleep apnea (OSA) is a significant risk factor for cardiovascular disease (CVD) with increasing prevalence. An important mechanism of CVD development is a dysregulation of the autonomic nervous system (ANS). This prospective and controlled cohort study aimed to investigate ANS function in OSA including the response to long-term continuous positive airway pressure (CPAP) therapy by analyzing 24 h Holter electrocardiogram and 24 h Holter ambulatory blood pressure recording parameters. The study enrolled 57 patients who were newly diagnosed with severe OSA. After 6 months of CPAP therapy, 37 patients had a good therapy adherence (usage of CPAP device >4 h per night), and their data were analyzed. The difference in nocturnal diastolic blood pressure values before and after CPAP therapy reached statistical significance (76 (68–84) vs. 74 (63–80) mmHg, *p* = 0.0439). Lower nocturnal values after CPAP therapy of SDNN (101.5 vs. 95 ms, *p* = 0.0492) and RMSSD (29.5 vs. 26 ms, *p* = 0.0193) were found. An increase in diurnal spectral power (1742 vs. 2112 ms^2^, *p* = 0.0282) and a decrease in nocturnal spectral power (3256 vs. 2124 ms^2^, *p* = 0.0097), nocturnal VLF band (2493 vs. 1485.4 ms^2^, *p* = 0.0176), nocturnal LF band (638.7 vs. 473 ms^2^, *p* = 0.0097), and nocturnal HF band (234.9 vs. 135.7 ms^2^, *p* = 0.0319) was found. The results showed an imbalance of the ANS with a sympathetic predominance, especially during the night hours and in those with arterial hypertension. The impact of CPAP therapy on the improvement in ANS parameters was more pronounced at night, in men, and those with arterial hypertension.

## 1. Introduction

Obstructive sleep apnea (OSA) is a disorder caused by a complete or partial obstruction of the upper airways during sleep [1]. The severity of the disease is determined by the apnea–hypopnea index (AHI), which indicates the number of apneas and hypopneas during one hour of sleep. The diagnosis of OSA is confirmed if the AHI is ≥5, a moderate degree of the disease is if the AHI is ≥15, and a severe degree if the AHI is ≥30. OSA syndrome (OSAS) is a condition in which there is daytime sleepiness with an AHI ≥ 5 [2].

OSA is still an underdiagnosed disorder, with an estimated prevalence of 936 million people worldwide [3]. Over the last two decades, the prevalence of OSA has been increasing, likely following the increase in obesity prevalence. Men are affected twice as often as women [4]. The severity of OSA is not related to the intensity of daytime sleepiness but to age, obesity, and arterial hypertension [5].

Arterial hypertension and OSA have an interdependent pathophysiology in terms of the same risk factors for both conditions, such as obesity. Both entities are risk factors for cardiovascular disease (CVD), and in addition, OSA is an independent risk factor for the development of arterial hypertension [6]. The prevalence of OSA in patients with hypertension, heart failure, coronary artery disease, pulmonary hypertension, atrial fibrillation, and stroke is between 40% and 80% [7]. A cross-sectional study showed that 82% of patients with resistant hypertension had OSA, and 55.5% had moderate or severe OSA with a higher nocturnal SBP and pulse pressure and a higher prevalence of nondipping pattern [8]. The pathophysiological mechanisms of the development of arterial hypertension in patients with OSA include a disorder of the autonomic nervous system (ANS) with excessive sympathetic activation [9]. The physiological drop in blood pressure during sleep (at least by 10% of the daily value—dipper status) is also reduced in patients with OSA [6]. A nondipper status is associated with an increased incidence of cardiovascular events, independent of the presence of diurnal hypertension [10].

When analyzing polysomnography study (PSG) parameters, patients with hypertension exhibit more respiratory arousals than normotensive patients resulting in sleep fragmentation [11]. Recurrent arousals lead to sympathetic activation, inflammation, oxidative stress, and metabolic dysfunction promoting the development of CVD [12]. Patients with hypertension have more severe OSA, a higher concentration of uric acid and electrolyte disbalance with a lower magnesium concentration and an altered calcium metabolism when compared with normotensive patients [11]. These changes can have a proarrhythmic effect and also participate in the development of CVD. 

Heart rate variability (HRV) is a term that describes fluctuations in the time intervals between two consecutive heartbeats (the so-called R-R interval). HRV parameters are a reflection of the interaction between the heart, the brain, and the ANS. The complex oscillations of the heart rhythm of a healthy heart enable a rapid adaptation of the cardiovascular system following physical and psychological needs [13]. Studies have shown an association between HRV and CVD such as myocardial infarction, stroke, angina, coronary artery disease (CAD), and sudden cardiac death. In the general population, decreased HRV values correlate with CAD and death [14].

The parameters of HRV can be divided by mathematical models into linear and nonlinear. The use of linear parameters is widespread. They are furthermore divided into time and frequency domains. Time-domain parameters of HRV include the standard deviation of all NN intervals (SDNN), the standard deviation of the averages of NN intervals in all 5 min segments of the entire recording (SDANN), the square root of the mean of the sum of the squares of differences between adjacent NN intervals (RMSSD), the mean of the standard deviations of all NN intervals for all 5 min segments of the entire recording (SDNN index), the number of pairs of adjacent NN intervals differing by more than 50 ms in the entire recording (NN50 count), and the NN50 count divided by the total number of all NN intervals (pNN50) [15].

The sympathetic and parasympathetic components of the ANS are represented in different parameters of the frequency domain which is based on the speed of their action. Sympathetic activation and inactivation are relatively slow and thereby affect the low-frequency range of the spectral analysis (low-frequency, LF) which is 0.04–0.15 Hz [15]. The parasympathetic activation and inactivation are faster and therefore are reflected in the area of higher frequencies of the spectral analysis (high-frequency, HF), 0.15–0.4 Hz [15]. The ratio of spectrum power in the low and high-frequency band (LF/HF ratio) is used as an indicator of sympathetic–parasympathetic balance [16]. Very low frequency (VLF) power is more associated with overall mortality than LF and HF power. A low VLF power is associated with arrhythmic mortality, post-traumatic stress disorder (PTSD), inflammation, and low testosterone. Physiological mechanisms that contribute to VLF power are not completely understood [13].

The dysregulation of the ANS in patients with OSA is one of the most important pathological mechanisms of the development of cardiovascular comorbidities. Studies have shown that repetitive airway obstruction, intermittent hypoxia, and hypercapnia lead to the activation of chemoreceptors that are responsible for excessive sympathetic activation [17]. Sympathetic activity is acutely elevated during episodes of apnea and hypopnea, which over time leads to a chronically elevated sympathetic tone throughout the day [17]. ANS alterations have a significant role in the development of hypertension, cardiac arrhythmias, and heart failure. Changes in the ANS in OSA patients also affect other organ systems causing altered gastric acid secretion, hypersecretion of adrenaline and noradrenaline, nocturia and hyperstimulated urinary bladder, and sexual dysfunction [17]. The analysis of a total of 12 studies showed an increased sympathetic activity and decreased parasympathetic activity in patients with OSA compared to a healthy population. The VLF power component was higher in subjects with OSA. Most studies showed higher values of LF power, LF/HF ratio, and lower values of HF power in patients with OSA, but the results were not unambiguous. Analyzing time-domain parameters, patients with OSA had lower values of SDNN, SDANN, and RMSSD, also with results that were not unambiguous in all studies [18]. 

A recent study showed that patients with OSA had an increased number of abnormalities in 24 h ECG Holter monitoring including more episodes of supraventricular tachycardia (SVT) and ventricular arrhythmias (VPC) with a positive correlation between AHI and average heart rate, supraventricular arrhythmias (SVPC) pairs, SVT, and pauses >2.5 s. Also, a higher AHI was an independent predicator for the increased number of pauses >2.5 s, SVT, and SVPC pairs [19].

The treatment of OSA consists of CPAP therapy, mandibular advancement device (MAD), or surgical procedures such as lateral pharyngoplasty techniques or uvulopalatopharyngoplasty [20]. Therapy with a CPAP device and MAD also can be used in patients with sleep bruxism (SB), a condition that affects 33–50% of patients with OSA, which was demonstrated in a recent study [21]. According to European Respiratory Society guidelines, CPAP therapy, compared to MAD, has a greater impact on an AHI decrease, independently of OSA severity, and a higher impact on a nocturnal SBP decrease in severe OSA [22].

Logically, the parameters describing the function of the ANS should be improved by eliminating the main cause of all pathological changes, namely, airway obstruction. According to some studies, CPAP therapy reduces acute and chronic sympathetic activity and improves nocturnal blood pressure in otherwise healthy individuals as well as those with CVD [23].

Although standard treatment of OSA with CPAP reduces the incidence of the primary pathophysiological event, large randomized trials have not confirmed a reduction in cardiovascular risk with the use of CPAP [24,25,26]. A recent prospective study showed that moderate and severe OSA is a risk factor for major adverse cardiovascular and cerebrovascular events (MACCE), but CPAP treatment was not associated with a lower rate of MACCE [27]. A meta-analysis showed that using CPAP for more than 4 h during the night led to a clinical and statistical reduction in MACCE [28]. These results suggest that treatment adherence could play a crucial role in reducing MACCE.

Previous studies have shown that there is ANS dysregulation in OSA that contributes to the development of CVD, but the pathophysiologic mechanisms are still not entirely clear. Not only are the results of studies investigating ANS and OSA different but also those on the effect of CPAP therapy on the ANS. Some studies explained unexpected results with a poor adherence to CPAP therapy and a short follow-up period, which was avoided in this study. The aim of this study was to investigate ANS function in OSA patients with a good CPAP therapy adherence.

## 2. Materials and Methods

### 2.1. Patient Selection and Study Design

This clinical study was prospective and controlled. The study was reported in accordance with the STROBE statement. Eligible patients were aged 18 years and above and had suspected OSA. Inclusion criteria was newly diagnosed severe OSA. Patients were recruited during the period from July 2019 to November 2020 after being diagnosed with severe OSA (AHI ≥ 30) based on an overnight hospital PSG performed with the following device: Respironics Alice^®^ 6: Sleepware G3 3.7.3, Philips, Murrysville, PA, USA. Results were interpreted according to American Academy of Sleep Medicine rules for scoring respiratory events [2]. All data were stored on a computer, manually scored, and evaluated by a certified sleep physician and technician. Apnea was considered as a complete cessation of breathing, i.e., a signal reduction by ≥90% for ≥10 s. Hypopnea was considered as a partial respiratory arrest, i.e., a signal reduction ≥30% lasting ≥10 s with a drop in oxygen saturation by ≥3% or arousal.

Exclusion criteria were a history of severe chronic obstructive pulmonary disease (COPD), severe chronic kidney disease (CKD), atrial fibrillation, acute heart failure, cerebrovascular insult or transient ischemic attack in the last 6 months, acute coronary syndrome in the last 6 months, and pregnancy. 

The study was approved by the local ethics committee. All patients signed a written consent after being informed of the nature and purpose of this study. After the first study visit, 57 patients were enrolled (1 patient who initially entered the study dropped out because of newly diagnosed atrial fibrillation). All measurements and recordings were performed before the patients started using continuous positive airway pressure (CPAP) therapy at night. CPAP devices from two manufacturers were prescribed (prisma Smart, Loewenstein Medical Technology GmbH + Co. KG, Hamburg, Germany; S9 Escape, ResMed, San Diego, CA, USA). CPAP treatment was administered during hospitalization, and the titration pressure assumed to be effective in treating the majority of events was defined at the 95% percentile. At the first study visit, the assessment included anthropometric measurements (height and weight), an Epworth sleepiness scale (ESS) questionnaire, medical history, a 24 h Holter electrocardiogram (Holter ECG) recording (performed on the Diagnostic Monitoring Software device—DMS300-4L, Cardionics, Bruxelles, Belgium) and processed with the program Cardioscan DM Software v. 12.5.0078a, Cardionics, Bruxelles, Belgium analyzing premature beats, arrhythmias, pauses, HRV in time and frequency domains, and QTc interval) and a 24 h Holter ambulatory blood pressure recording (performed on the device Ambulatory Blood Pressure Monitor—model Oscar 2B/222B, SunTech Medical, Morrisville, NC, USA and processed by the program MGY-ABP1 monitor v. 1.1.0031a, SunTech Medical, Morrisville, NC, USA, analyzing overall, diurnal, nocturnal, and maximal values of SBP and DBP and dipper status). The follow-up study visit was after a minimum of 6 months of CPAP device usage and included the same assessment as the first visit. Of 57 enrolled patients, 37 had a good CPAP adherence (usage of the device for at least 4 h per night, data obtained from the device’s memory card) and their data were evaluated in this study. This study is an extension of a previously published study, using the same group of patients [29].

The study was conducted in accordance with the Declaration of Helsinki and approved by the Ethics Committees of University Psychiatric Hospital Vrapče (23–574/1–18, 12 February 2018) and Clinical Hospital Sveti Duh (01-388, 25 January 2018).

### 2.2. Statistical Analysis

Data are presented in tables and graphics. The normality of the data distribution was tested using the Shapiro–Wilk test, and corresponding statistical methods were applied according to the obtained results. Continuous variables were expressed as means and standard deviations (normal distribution) or medians and interquartile ranges (non-normal distribution). Categorical variables were expressed as numbers (percentages). Differences in variables were tested using a paired-samples *t*-test or Wilcoxon test, depending on the data normality. Corresponding correlation coefficients between the observed clinical variables were obtained using Pearson’s correlation coefficient and the Mann–Whitney U test. All *p* values < 0.05 were considered significant. Python version 3.8 programming language was used in the analysis.

## 3. Results

After the follow-up period of 6 months the data of 37 patients who had satisfactory CPAP therapy adherence were analyzed. Demographic and clinical characteristics of the patients are summarized in Table 1. 

The mean age was 53 ± 10 years. The majority of patients were male (78%), obese (average BMI was 34.4 ± 6.1 kg/m^2^), and 27% of them were smokers. The most common comorbidity was arterial hypertension (51%), followed by diabetes (16%). Patients generally did not have previous cardiovascular diseases, only one patient had acute heart failure more than 6 months before inclusion in the study. Also, only two patients had a mild degree of COPD.

The polysomnographic parameters, assessment of sleepiness, and duration of CPAP therapy are shown in Table 2. 

The average AHI of the study group was 58.4 ± 22. Polysomnography confirmed a predominantly obstructive component in apnea episodes (OA 33.35 vs. CA 1.3). The average duration of apnea was 24.2 ± 6.1 seconds, and the average lowest measured oxygen saturation during sleep was 74.03 ± 11.36%. The mean score of the Epworth sleepiness scale was 10.6 ± 5.2. The mean duration of CPAP therapy was 290.49 ± 56.7 days. The average time of CPAP therapy during the night was (322.3 ± 51.3 min) and the AHI after therapy was 4 (2.9–5).

The Holter ambulatory blood pressure recording parameters before and after CPAP therapy are shown in Table 3.

From all the parameters recorded, only the difference in nocturnal diastolic blood pressure values before and after CPAP therapy reached statistical significance (76 (68–84) vs. 74 (63–80) mmHg, *p* = 0.0439).

Figure 1, Figure 2 and Figure 3 show the statistically significant correlations between Holter ambulatory blood pressure recording parameters. The correlation coefficients and *p*-values are shown in the figures.

Holter ECG recording parameters, HRV and spectral analysis are shown in Table 4. 

The average heart rate was lower after CPAP therapy (78 (73–82) vs. 74 (69–79) beats/min, *p* = 0.0165). In the HRV time domain, statistically significant lower nocturnal values before and after CPAP therapy of SDNN (101.5 (88–122) vs. 95 (73.75–108) ms, *p* = 0.0492) and RMSSD (29.5 (22.25–39.75) vs. 26 (20.5–31.75) ms, *p* = 0.0193) were found. In the frequency domain, a statistically significant increase in diurnal spectral power (1742 (1205–2794) vs. 2112 (1457–2721) ms^2^, *p* = 0.0282) and a decrease in nocturnal spectral power (3256 (1923–5603) vs. 2124 (1544–4129) ms^2^, *p* = 0.0097), nocturnal VLF band (2493 (1348–3735) vs. 1485.4 (1072–2727) ms^2^, *p* = 0.0176), nocturnal LF band (638.7 (377.85–1273.85) vs. 473 (315.5–949.95) ms^2^, *p* = 0.0097) and nocturnal HF band (234.9 (118.95–360.2) vs. 135.7 (98–286.1) ms^2^, *p* = 0.0319) were found. 

Figure 4, Figure 5 and Figure 6 show the statistically significant correlations between Holter ECG recording parameters, including heart rate variability parameters and Holter ambulatory blood pressure recording parameters. The correlation coefficients and *p*-values are shown in the figures.

## 4. Discussion

The impact of CPAP therapy on blood pressure values has been evaluated in many studies. In a meta-analysis of 31 randomized clinical studies, the data obtained by a continuous measurement of arterial pressure over 24 h supported a reduction in SBP by 2.2 ± 0.7 mmHg during the day and 3.8 ± 0.8 mmHg during the night, and in DBP by 1.9 ± 0.6 mmHg during the day and 1.8 ± 0.6 during the night. The decrease in SBP was proportional to AHI [30].

This study, analyzing the data before and after CPAP therapy, showed a reduction in the average values of systolic and diastolic blood pressures, in total and separately during the day and night periods, but only the reduction in the nocturnal DBP value reached statistical significance. Also, a greater drop in nocturnal systolic and diastolic pressures was observed after CPAP therapy but without reaching statistical significance (Table 3). The results of this study are accordant with previously published data [31]. The failure to achieve statistical significance in the measurements of most Holter blood pressure parameters can be explained by the small sample, but also by the fact that 19 patients were already taking antihypertensive drugs before starting CPAP therapy, and that the antihypertensive therapy was changed between the two study visits in 9 patients (in 8 patients, the dose of antihypertensive drug was increased and in 1 patient, it was decreased).

A further analysis of the correlations revealed statistically significantly higher values of nocturnal DBP in men (Figure 1) before CPAP therapy. After CPAP therapy, the change in nocturnal DBP was more pronounced in women and patients who did not have arterial hypertension (Figure 2). A greater change in nocturnal DBP values was accompanied by a greater change in nocturnal SBP and diurnal systolic and diastolic pressure (Figure 3).

The effect of CPAP therapy on blood pressure values is more pronounced during the night, as determined by a recent meta-analysis of eight randomized clinical studies on 606 patients. A possible explanation for this phenomenon is sympathetic activation, which is more pronounced in patients with OSA during the night, and CPAP acts primarily on the nocturnal component of sympathetic activation [31]. The significant correlations between changes in nocturnal DBP and nocturnal SBP and diurnal systolic and diastolic pressure could potentially be explained with the assumption that changes in nocturnal DBP are the initial changes that occur during CPAP therapy, and that they are a possible predictor of a further decrease in blood pressure values. The fact that from all measured parameters, only the values of nocturnal DBP achieved a statistically significant decrease could also be explained by results from previous studies that had shown that CPAP treatment was more effective in resistant hypertension compared to nonresistant hypertension, given that sympathetic activation and changes in vasculature reactivity in OSA were more pronounced in patients with resistant hypertension [32].

Analyzing the data of Holter ECG recordings for 24 h, a statistically significant difference in the average heart rate before and after CPAP therapy was obtained (Table 4).

The HRV time-domain parameters and nocturnal SDNN and RMSSD values were statistically significantly lower after the CPAP therapy (Table 4.). Higher values of the parameters of the time domain indicate a better adaptation capability of the ANS. In most studies, the values of time-domain parameters were decreased in patients with OSA compared with healthy subjects [18]. However, a study by Kim et al. showed elevated values of SDNN and RMSSD in patients with OSA [33]. Therefore, previous findings on the dynamics of the time-domain parameters would assume an increase in their values after CPAP therapy. This paradoxical result may be because time-domain parameters are affected by total HRV, and a more detailed analysis of the sympathetic/parasympathetic imbalance is not possible. In this study, the nocturnal values of SDNN were lower in subjects with arterial hypertension (Figure 4), and the nocturnal values of RMSSD and SDNN were lower in patients with higher values of nocturnal SBP (Figure 5). The aforementioned correlations speak in favor of a reduced adaptability of the ANS during the night h in patients with OSA and arterial hypertension.

Among diurnal parameters of the frequency domain, the total power showed a statistically significant increase after CPAP therapy (Table 4). Most of the total power value comes from the VLF band (power), the dynamics of which are still not sufficiently explored. Previous findings have shown significantly increased values of the VLF band in patients with OSA [33].

Values of nocturnal total power and VLF band showed a statistically significant decrease (Table 4), which is consistent with the dynamics of these parameters obtained by previous research [33]. Diurnal and nocturnal total spectral power was significantly lower in patients with arterial hypertension (Figure 4). Nocturnal values of the VLF band were also lower in patients with arterial hypertension (Figure 4). Nocturnal values of total power were significantly lower in patients with higher values of nocturnal SBP (Figure 5). Diurnal values of total spectral power were lower at higher AHI values (Figure 5). These correlations showed lower spectral power values in patients with arterial hypertension and a more severe degree of OSA, which could not be explained by the ANS imbalance alone.

A greater decrease in the value of the nocturnal spectral power was recorded in men and patients with arterial hypertension (Figure 6). Also, a greater drop in diurnal spectral power was recorded in patients with arterial hypertension (Figure 6). The mentioned significant correlations speak in favor of a greater impact of CPAP therapy on the improvement in HRV parameters in men and patients with arterial hypertension.

The values of the LF band decreased during the day and night, but only the measurements during the night period showed statistical significance (Table 4). The LF band consists mostly of frequencies coming from the sympathetic activity, and its values are mostly elevated in patients with OSA [18]. The results obtained in this study speak in favor of a reduction in the activity of the sympathetic component of the ANS after the use of CPAP therapy, which was also confirmed by other authors [27]. The values of the LF band during the night before CPAP therapy were higher in younger patients and those with lower blood pressure values (Figure 5), which speaks in favor of a greater sympathetic activation in those patients. The decrease in the values of the LF band after CPAP therapy was greater in men and patients with arterial hypertension (Figure 6), which can be explained by a possible better suppression of the sympathetic activity by CPAP in these patients.

The values of the HF band during the night decreased statistically significantly (Table 4) and was even more pronounced in patients with arterial hypertension (Figure 6). In the majority of studies published so far, the values of the HF band are decreased in OSA. Nastalek et al. showed a statistically significant increase in the value of the HF area after CPAP therapy [34]. Nocturnal values of the HF band were higher in younger patients and those with lower values of nocturnal SBP (Figure 5). Recent research has shown an association between parasympathetic activity and long-term cardiovascular outcomes. Namely, the HF band was shown to be an independent predictor of cardiovascular outcomes with a positive correlation with latency for the development of CVD, which the authors interpret as a possible decrease in parasympathetic activity years before the occurrence of a cardiovascular event [14]. The LF/HF ratio did not change significantly before and after CPAP therapy (Table 4).

A frequency-domain analysis in OSA patients can be challenging due to the presence of repetitive apnea episodes, arousals, and leg movements, which induce biologic noise and can modify the findings [35]. According to some authors, nonlinear HRV parameters may be superior for assessing autonomic cardiovascular regulation in OSA patients because the mechanisms involved in cardiovascular regulation occur in a nonlinear fashion over a long time period, and they are less affected by artifacts [36]. Results of a recent study show that the dysregulation of HRV with an overactivation of sympathetic tone could even be associated with cerebral white matter changes in patients with OSA [36].

There is a known association between OSA and sudden cardiac death during the night hours [37]. The QT interval is an electrocardiographic correlate of ventricular depolarization and repolarization, including the period susceptible to the occurrence of circular tachycardia. Research has linked the prolongation of the QT interval with an increased risk of malignant arrhythmias and sudden cardiac death [37]. Research by Rossi et al. showed a prolongation of the QTc interval in patients with OSA who were already using CPAP therapy. After stopping the use of the therapy for two weeks, the prolongation of the QTc interval was greater compared to the initial values in comparison with patients who continued to use CPAP [38]. However, a study by Viigimae et al. showed no significant difference in QTc interval values in patients with OSA compared to patients without OSA, but the QT interval variability index was significantly higher in patients with OSA [39]. QT/QTc values showed no significant change before and after CPAP therapy in this study (Table 4). It is possible that some other parameters describing the QT Interval could explain the imbalance of ventricular de-/repolarization in patients with OSA.

One of the limitations of this study is the small sample number. In addition, the sample does not represent the usual characteristics of the general population. The patients were predominately male and had a relatively small number of comorbidities. In addition, the patients had individualized chronic therapy (including antihypertensive agents and beta-blockers), some of which were changed during follow-up. Future research in this area could provide new insights if data from a larger and more representative cohort or a randomized control trial with a longer follow-up period were used. The effects of CPAP settings and genotype in OSA patients in relation to ANS dysfunction and cardiovascular comorbidities also need to be explored. 

## 5. Conclusions

The analysis of the data in this study showed an imbalance of the ANS with a predominance of the sympathetic nervous system in patients with OSA, especially during the night hours and in those with arterial hypertension. The impact of CPAP therapy on the improvement in heart variability parameters is more pronounced at night, in men, and in patients with arterial hypertension.

## Figures and Tables

**Figure 1 biomedicines-11-03210-f001:**
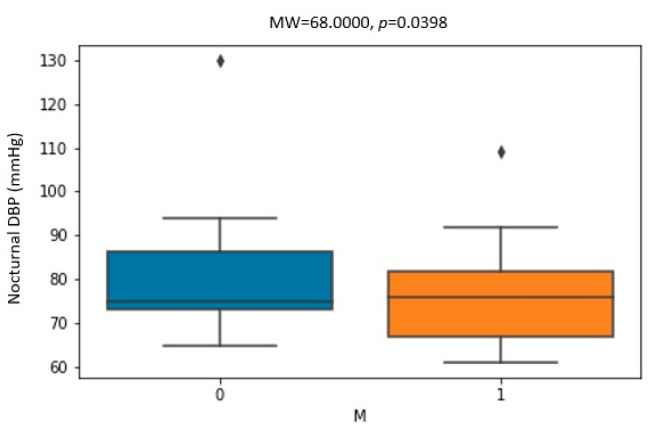
Correlation between nocturnal diastolic blood pressure (DBP) and gender (M—male) before CPAP therapy. Dots placed past the line edges indicate outliers.

**Figure 2 biomedicines-11-03210-f002:**
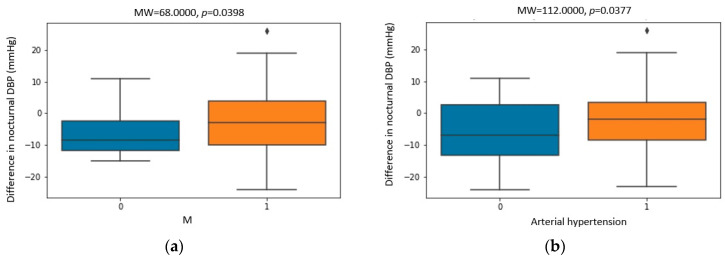
Correlation between (**a**) the difference in nocturnal diastolic blood pressure (DBP) before and after CPAP therapy and gender (M—male) and (**b**) the difference in nocturnal diastolic blood pressure (DBP) before and after CPAP therapy and arterial hypertension. Dots placed past the line edges indicate outliers

**Figure 3 biomedicines-11-03210-f003:**
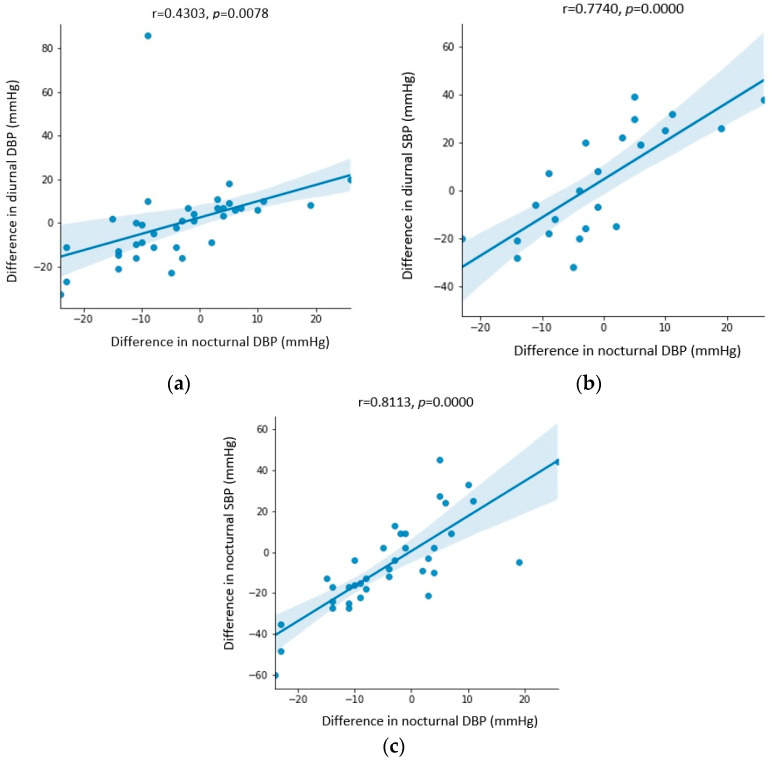
Correlation between (**a**) the difference in diurnal diastolic blood pressure (DBP) before and after CPAP therapy and the difference in nocturnal diastolic blood pressure (DBP) before and after CPAP therapy, (**b**) the difference in diurnal systolic blood pressure (SBP) before and after CPAP therapy and the difference in nocturnal diastolic blood pressure (DBP) before and after CPAP therapy, and (**c**) the difference in nocturnal systolic blood pressure (SBP) before and after CPAP therapy and the difference in nocturnal diastolic blood pressure (DBP) before and after CPAP therapy.

**Figure 4 biomedicines-11-03210-f004:**
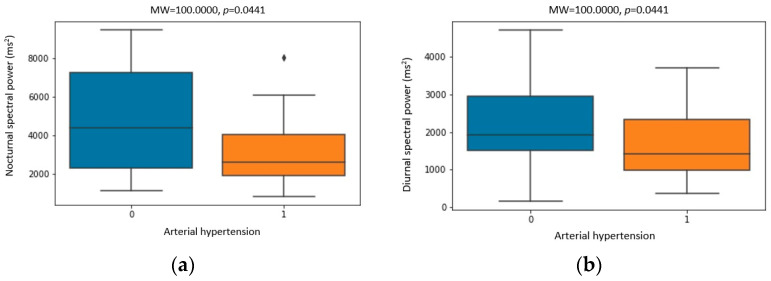
Correlation between arterial hypertension and (**a**) nocturnal spectral power (ms^2^), (**b**) diurnal spectral power (ms^2^), (**c**) nocturnal VLF power (ms^2^) and (**d**) nocturnal SDNN (ms). All parameters were recorded before CPAP therapy. Dots placed past the line edges indicate outliers

**Figure 5 biomedicines-11-03210-f005:**
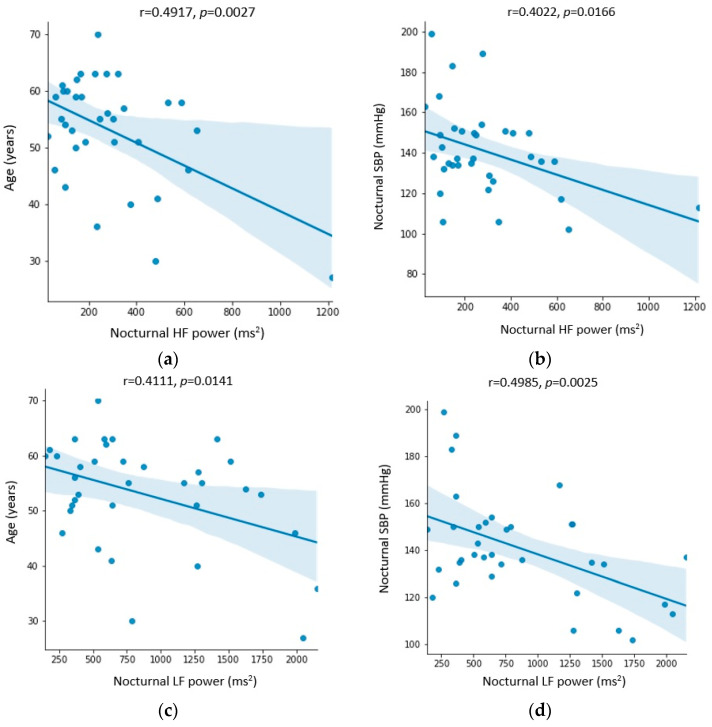
Correlation between (**a**) age and nocturnal HF power (ms^2^), (**b**) nocturnal systolic blood pressure (SBP) (mmHg) and nocturnal HF power (ms^2^), (**c**) age and nocturnal LF power (ms^2^), (**d**) nocturnal systolic blood pressure (SBP) (mmHg) and nocturnal LF power (ms^2^), (**e**) nocturnal systolic blood pressure (SBP) (mmHg) and nocturnal spectral power (ms^2^), (**f**) nocturnal systolic blood pressure (SBP) (mmHg) and nocturnal RMSSD (ms), (**g**) nocturnal systolic blood pressure (SBP) (mmHg) and nocturnal SDNN (ms), and (**h**) apnea–hypopnea index (AHI) and diurnal spectral power (ms^2^). All parameters were recorded before CPAP therapy.

**Figure 6 biomedicines-11-03210-f006:**
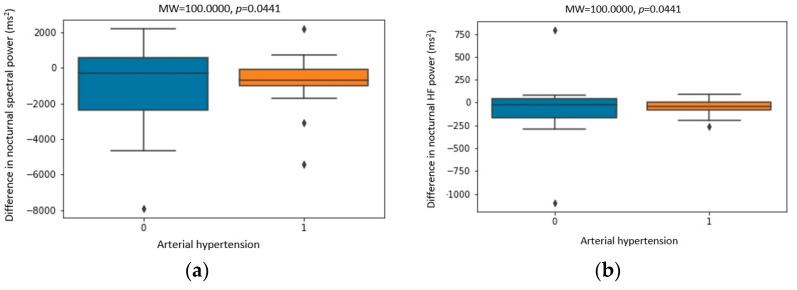
Correlation between (**a**) the difference in nocturnal spectral power (ms^2^) before and after CPAP therapy and arterial hypertension, (**b**) the difference in nocturnal HF power (ms^2^) before and after CPAP therapy and arterial hypertension, (**c**) the difference in nocturnal LF power (ms^2^) before and after CPAP therapy and arterial hypertension, (**d**) the difference in nocturnal LF power (ms^2^) before and after CPAP therapy and gender (M—male), (**e**) the difference in nocturnal spectral power (ms^2^) before and after CPAP therapy and gender (M—male), and (**f**) the difference in diurnal spectral power (ms^2^) before and after CPAP therapy and arterial hypertension. Dots placed past the line edges indicate outliers.

**Table 1 biomedicines-11-03210-t001:** Demographic and clinical characteristics of the study group.

Age (years)	53 ± 10
Male/female gender	29/8
BMI (kg/m^2^)	34.4 ± 6.1
Smoker (number/%)	10/27
Comorbidities	
Hypertension (number/%)	19/51
CVD	1 *
COPD	2
Diabetes mellitus (number/%)	6/16
CKD	1 *

BMI—body mass index, CKD—chronic kidney disease, CVD—cardiovascular disease including acute coronary syndrome, stroke/transient ischemic attack/heart failure, COPD—chronic obstructive pulmonary disease. * Only one patient had a history of heart failure and mild CKD.

**Table 2 biomedicines-11-03210-t002:** Polysomnographic parameters, assessment of sleepiness, and duration of CPAP therapy of the study group.

Polysomnographic parameters	
AHI (events/h)	58.4 ± 22
CA (events/h)	1.3
OA (events/h)	33.35
Hypopnea (events/h)	11.15
Average apnea duration (s)	24.2 ± 6.1
Minimum O^2^ saturation (%)	74.03 ± 11.36
ESS	10.6 ± 5.2
Average time of CPAP therapy per night (min)	322.3 ± 51.3
Duration of CPAP therapy (days)	290.49 ± 56.7

AHI—apnea–hypopnea index, CA—central apnea, CPAP—continuous positive airway pressure, ESS—Epworth sleepiness scale, OA—obstructive apnea.

**Table 3 biomedicines-11-03210-t003:** Twenty-four-hour ambulatory blood pressure monitoring parameters.

	Before CPAP Therapy	After CPAP Therapy	*p*-Value
Systolic blood pressure			
Diurnal (mmHg)	147 (131–156)	139 (131–156)	0.7833
Nocturnal (mmHg)	137 (129–150)	130 (117–146)	0.0853
24 h (mmHg)	145 (128–153)	139 (125–154)	0.3814
Maximum value (mmHg)	176.81 ± 26.35	177.43 ± 22.85	0.9027
Diastolic blood pressure			
Diurnal (mmHg)	82 (76–90)	80 (73–89)	0.4319
Nocturnal (mmHg)	76 (68–84)	74 (63–80)	0.0439
24 h (mmHg)	81 (73–88)	78 (72–85)	0.141
Maximum value (mmHg)	107 (96–118)	107 (96–119)	0.8621
SBP dipping (mmHg)	4.57 ± 7.97	7.31 ± 7.67	0.0706
DBP dipping (mmHg)	9.3 (2.9–15.8)	10.5 (4.2–18.7)	0.1818

DBP—diastolic blood pressure, SBP—systolic blood pressure.

**Table 4 biomedicines-11-03210-t004:** Holter ECG recording parameters, including heart rate variability parameters.

	Before CPAP Therapy	After CPAP Therapy	*p*-Value
Minimum HR (/min)	49.08 ± 6.09	49.73 ± 5.59	0.5013
Maximum HR (/min)	129 (118–140)	124 (116–137)	0.8937
Average HR (/min)	78 (73–82)	74 (69–79)	0.0165
SVEB (number)	12 (3–60)	7 (2–46)	0.5437
VEB (number)	5 (1–72)	4 (0–50)	0.1629
VT (1—yes, 0—no)	0	0	
Pauses >2 s (1—yes, 0—no)	0	0	
Time domain			
SDNN-24 h (ms)	128.25 ± 31.79	128.61 ± 30.24	0.9396
SDANN index (ms)	120 (97.5–142)	109 (96.5–137)	0.8844
SDNN index (ms)	47 (39–56.5)	44 (39–53.5)	0.9715
RMSSD-24 h (ms)	23 (18–27)	20 (18–26.5)	0.3632
pNN50-24 h (%)	3 (1–6)	3 (1–5.5)	0.2529
SDNN day (ms)	103.53 ± 25.84	104.12 ± 25.81	0.8738
RMSSD day (ms)	17.5 (15–21.75)	18 (15–23)	0.6396
pNN50 day (%)	1 (0–2.75)	1 (0–4)	0.8641
SDNN night (ms)	101.5 (88–122)	95 (73.75–108)	0.0492
RMSSD night (ms)	29.5 (22.25–39.75)	26 (20.5–31.75)	0.0193
pNN50 night (%)	7 (3–15.25)	6 (2–10.75)	0.0892
Frequency domain			
Spectral power average (ms^2^)	2470 (1529–3754)	2187 (1636–3203)	0.9499
VLF average (ms^2^)	1855 (1138–2911)	1677 (1116–2294)	0.7432
LF average (ms^2^)	508 (289.35–76.8)	393.9 (294.6–685.6)	0.359
HF average (ms^2^)	155.6 (71.25–219.9)	92.6 (71.75–175.45)	0.1318
Spectral power day (ms^2^)	1742 (1205–2794)	2112 (1457–2721)	0.0282
VLF day (ms^2^)	1328 (972–2083)	1586 (1069–1989)	0.0513
LF day (ms^2^)	378.5 (204.85–627.2)	396 (220.4–625.9)	0.3339
HF day (ms^2^)	69.8 (37.95–123.6)	66.6 (44.25–125.75)	0.7432
Spectral power night (ms^2^)	3256 (1923–5603)	2124 (1544–4129)	0.0097
VLF night (ms^2^)	2493 (1348–3735)	1485 (1072–2727)	0.0176
LF night (ms^2^)	638.7 (377.85–1273.85)	473 (315.50–949.95)	0.0097
HF night (ms^2^)	234.9 (118.95–360.2)	135.7 (98.00–286.1)	0.0319
LF/HF average	3.18 (2.36–5.92)	4.03 (2.73–5.03)	0.5445
LF/HF day	4.98 (3.14–6.56)	4.63 (3.26–6.47)	0.9608
LF/HF night	3.08 (1.85–5.12)	3.14 (2.06–4.73)	0.7186
QT (ms)	465 (444–483)	456 (442–480)	0.7063
QTc (ms)	492 (481–539)	499 (481.5–509)	0.3898

HF—absolute power of the high-frequency band (0.15–0.4 Hz), HR—heart rate, LF—absolute power of the low-frequency band (0.04–0.15 Hz), pNN50—percentage of successive RR intervals that differ by more than 50 ms, RMSSD—root mean square of the successive differences, SDANN index—mean of the standard deviations of the average NN intervals, SDNN—standard deviation of the NN (R-R) intervals, SVEB—supraventricular ectopic beat, VEB—ventricular ectopic beat, VLF—absolute power of the very low frequency band (0.0033–0.04 Hz), VT—ventricular tachycardia.

## Data Availability

The data presented in this study are available on request from the corresponding author. The data are not publicly available due to privacy restrictions.

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
