# Peer review of "Impact of CPAP Therapy on the Autonomic Nervous System"

_biomedicines, 2023, doi:10.3390/biomedicines11123210_

Round 1
Reviewer 1 Report
Comments and Suggestions for Authors
The topic of the study is significant and novel. However I found a few major flaws:
1. The study has to be report in accordance with the STROBE Statement https://www.equator-network.org/reporting-guidelines/strobe/ Please use the STROBE checklist.
2. Authors have to precisely describe what method was obstructive sleep apnea diagnosed and what devices and software were used for it?
3. Please define inclusion criteria for study participants.
4. Authors have to add a detail information about CPAP devices used in the study (brand name, manufacturer, country of origin) and details about all devices and software used in the study.
5. Please do not cite articles older than 10 years because mostly they are outdated. Authors should consider to cite the latest and reliable articles strongly related to the topic:
- Urbanik D, Gać P, Martynowicz H, Podgórski M, Poręba M, Mazur G, Poręba R. Obstructive Sleep Apnea as a Predictor of Arrhythmias in 24-h ECG Holter Monitoring. Brain Sci. 2021 Apr 12;11(4):486. doi: 10.3390/brainsci11040486.
This latest study proofed that higher AHI constituted an independent predicator for the increased number of pauses >2.5 s, SVT, and SVPC pairs in 24-h ECG Holter monitoring. In summary, patients with OSA are characterized by the increased number of abnormalities in 24-h ECG Holter monitoring. Authors should present it in Introduction and/or discuss it.
- Martynowicz H, Wieczorek T, Macek P, Wojakowska A, Poręba R, Gać P, Mazur G, Skomro R, Smardz J, Więckiewicz M. The effect of continuous positive airway pressure and mandibular advancement device on sleep bruxism intensity in obstructive sleep apnea patients. Chron Respir Dis. 2022 Jan-Dec;19:14799731211052301. doi: 10.1177/14799731211052301.
This latest study proofed that CPAP and MAD treatments were effective against OSA and SB. Authors should present it in Introduction and/or discuss it.
- Kanclerska J, Wieckiewicz M, Nowacki D, et al. Sleep architecture and vitamin D in hypertensives with obstructive sleep apnea: A polysomnographic study [published online as ahead of print on October 20, 2023]. Dent Med Probl. doi:10.17219/dmp/172243
This study presents the latest findings related to OSA etiology. Authors should present it in Introduction.
Comments on the Quality of English Language
The manuscript needs only minor language revision.
Reviewer 2 Report
Comments and Suggestions for Authors
Introduction
- Provide more background on the prevalence of obstructive sleep apnea (OSA) and associated cardiovascular risks and how OSA treatment reduce the risk. Cite doi:10.1111/eci.12908. and doi:10.1007/s11325-021-02520-y- Explain the role of autonomic dysfunction in the pathogenesis of cardiovascular disease in OSA patients. cite doi:10.1016/j.autneu.2019.102563.
- Discuss prior research on effects of OSA on sympathetic/parasympathetic balance and heart rate variability. cite doi:10.1152/ajpheart.00590.2021.
- Introduce continuous positive airway pressure (CPAP) therapy, its cardiovascular benefits and variability in individual responses. cite doi:10.1183/16000617.0200-2021
- Clearly state the rationale and objectives of your study analyzing impact of CPAP on autonomic function using holter monitoring.
Methods
- Provide more details on patient recruitment process, inclusion/exclusion criteria.
- Describe the polysomnography protocol and CPAP titration process.
- Explain holter ECG and ABPM monitoring methodology, equipment used, parameters analyzed.
- Give more information on statistical analysis - tests used for specific comparisons and correlations.
Results
- Report demographic and clinical characteristics using summary statistics like means/SD for continuous variables, percentages for categorical variables.
- Include tables summarizing pre/post CPAP changes in BP, HRV time-domain, frequency-domain and other ECG parameters. Comment on significance.
- When describing correlations, provide correlation coefficients and exact p-values for each association tested.
Discussion
- Compare your HRV results to previous studies analyzing impact of CPAP therapy on autonomic function. cite PMID: 17926981.
- Discuss possible mechanisms for CPAP effects on nocturnal BP and HRV metrics. cite doi:10.1016/j.atherosclerosis.2014.03.034.
- Acknowledge limitations like small sample size, demographic factors, medications use.
- Suggest future research directions - e.g. larger RCTs, role of CPAP settings, genotype effects.
Round 2
Reviewer 1 Report
Comments and Suggestions for Authors
I don't have further comments. The manuscript has been revised correctly.